# OpenReview forum: "Neologism Learning for Controllability and Self-Verbalization"
_ICLR.cc/2026/Conference — ICLR 2026 Poster_

### Official Review · Reviewer_zfub · 2025-10-31

**Soundness:** 3
**Presentation:** 3
**Contribution:** 3
**Rating:** 6
**Confidence:** 4

**Summary:**

This paper proposes neologism learning, a parameter-efficient method for controlling LLM behavior. The method involves adding a new token embedding to a frozen LLM and training only this new embedding on examples of a target concept. The paper introduces two concepts for interpretability: 1) self-verbalization, where the model can provide natural language descriptions (like synonyms) for its newly learned neologism. 2) plug-in evaluation, a method to validate these verbalizations by plugging them back into prompts and measuring if they control the target concept.

**Strengths:**

* The core idea of training the neologism using a parameter-efficient method is interesting and practical.
* Metrics such as self-verbalization and plug-in evaluation are a simple, effective, and easily reproducible method for validating the method in interpretability research.
* The discovery of "machine-only synonyms" (e.g., "lack" for brevity) is interesting and opens up a new line of research.

**Weaknesses:**

### 1. Data
The paper proves its method works on clean prompts (LIMA) to produce clean synthetic outputs (Gemini-style). How about more practical, messy, real-world prompts?

### 2. Model
The primary experiments are on Gemma-3-4B-IT, a relatively small model. While the "lack" example anecdotally transfers to Gemini-2.5-Flash, the paper lacks a systematic study of how neologism learning, and especially self-verbalization, scales with model size and architecture

### 3. Metric
To support the claim of controllability, metrics measuring some side effects would be valuable. (e.g., when they use flattery-word, maybe this makes response longer or harmful).

**Questions:**

* The paper shows learning 3 neologisms. What happens when you try to learn 100? Does the model's base knowledge degrade?
* If you train two different neologisms (from different random or word-based initializations) on the exact same dataset of a concept, do they converge to similar embedding vectors? More importantly, do they produce the same self-verbalizations?
* The paper compares against a few-shot ICL, but a more direct comparison to other parameter-efficient steering methods would be interesting.

---

> ### Author Response · Authors · 2025-11-20
>
> Thank you, reviewer zfub, for your review - we appreciate your assessment of our approach as “interesting”, “practical”, “easily reproducible” and the discovery of machine-unique synonyms as “opens up a new line of research”. Here are some thoughts to consider.
> - While we appreciate the need to validate methods on real-world data, LIMA questions are naturalistic and cover a broad range of topics, from summarizing an article about breach of copyright to asking whether 1021 is prime to discussing “death” without using the word death or euphemisms for death.
> - While testing neologism learning on a range of model sizes would be valuable, we instead opted for a huge range of experimental settings and ablations– testing different templates, testing different loss functions and regularizations, introducing self-verbalization, testing compositions of concepts either trained for composition (Section 6) or not trained for composition (Appendix A.6.) We do have a short experiment testing in-context definition of neologisms (naming concepts in few-shot examples without gradient descent) across Gemma, Gemini-2.5-Flash, and Gemini-2.5-pro  (Table 16).
> - Regarding “side-effects” of concept control, we agree, and we chose in particular the three concepts in Section 6 – short length, numerical, and likely-under-Gemini – to be in tension with each other in order to measure side effects more precisely. That is, answers that are shorter tend to have fewer numbers, answers with more numbers tend to be longer, and answers that are likely under gemini may be unlikely to be unusual in their length or number count. Indeed, we do see side effects in both our few-shot baseline and neologism learning, though fascinatingly, despite “short” and “likely under gemini” ending up correlated with each other, we find that responses generated from the composition of the “numerical” and “likely under gemini” concepts do not have the side effect of being shorter than usual, which we found exciting.
>
> As for answers to your specific questions:
> - Learning 100 neologisms – great question, and we do not know. However, unlike finetuning a model’s weights entirely, one nice feature of neologism learning is that whenever one does not use (any subset of) the neologisms, we can guarantee that we sample from the original unmodified model, so one can choose which “model” to pick (from which neologisms to use) independently for each element of a batch (unlike for LoRA). As a result, it might be possible to build up a “neologism library” over time where concepts that the community finds useful for steering model behavior are encoded as new tokens, and users can simply use the tokens they require for their own purpose.

---

> > ### Comment · Reviewer_zfub · 2025-11-25
> >
> > I thank the authors for their response, but I will maintain my score as the rebuttal did not fully address my concerns.

---

### Official Review · Reviewer_8hG7 · 2025-10-31

**Soundness:** 2
**Presentation:** 3
**Contribution:** 2
**Rating:** 2
**Confidence:** 4

**Summary:**

The authors propose Neologism Learning, a method that incorporates new concepts (i.e., neologisms) into a Large Language Model (LLM) solely focusing on the embedding projections,  that is without updating the parameters of the model. The authors suggest that, once trained, specific concepts such as “give a short answer”, “give a flattering answer”, or compositions thereof, placed in a prompt (e.g., at the end of a sentence like “describe the recipe of a lasagna”), allows to “control” the model’s behavior in a somewhat reliable way. The authors further suggest that the model can perform what they call “self-verbalization”, i.e.,  explain back (e.g., via listing a bunch of synonyms) the meaning of these neologisms. Finally, they observe that some of these synonyms work as machine-only concepts: they modify the model’s behavior even though they are meaningless to humans.

**Strengths:**

1. The paper is clearly written and well structured.

2. Although the core methods that give rise to the main method of this work are not novel, I find the overarching goal, i.e., Neologism Learning, to be a fresh perspective on LLM evaluation and output controllability.

**Weaknesses:**

1. I do not believe that the baselines proposed by the authors are fair. Given that the authors use “neologisms” related to concepts that already have a clear semantic meaning (e.g., give a short answer), it seems that a proper comparison would be to compare the model’s behavior output with a prompt incorporating a semantic description of the concept.

2. It is unclear whether we can consider the proposed concepts as “neologisms”. The latter are associated with (intrinsically) novel societal concepts, like the one provided by the authors in their paper (i.e., doomscrolling). In contrast, it seems that this work merely proposes novel bindings, a “pseudo-word” to an already existing meaning (via optimizing the NLL of a “bundle” of sentences that relate to an already existing concept). Hence, rather than learning a novel concept, LLMs may simply be learning to associate already encoded specific sequential token patterns to a novel input token. In fact, the authors at some point discuss out-of-distribution generalization which I think is a gross overstatement of the results presented in this work.

3. In fact, in light of my second point, the results of the plug-in evaluation method should not come as a surprise. The objective function pushes the novel (learned) concepts to be semantically associated (as in a knowledge graph) with words that relate new concepts’ description (or sentences it relates to). Thus, the results of the plug-in evaluation emerge from the ability of these models to perform 1-hop associative bindings of concepts and their semantic descriptions. In sum, these results would have been much more robust and convincing if the authors had focused on associating new tokens, to completely new concepts (such as doomscrolling).

**Questions:**

1. Have the authors compared the model's output to a non-default response, i.e. incorporating in the prompt instructions for the proper behavior with the semanctic description of the concept itself.

2. Have the authors thought of building completely novel concepts, that is related to semantic descriptions that may even be considered as unreal (or surreal).

---

> ### Author Response · Authors · 2025-11-20
>
> We thank reviewer 8hG7 for their effort and review. We’re glad to hear you found the paper “clearly written”, “well-structured” and “a fresh perspective on LLM evaluation and output controllability”.
>
>
> Here are our thoughts:
> - We do indeed compare neologism learning to a prompt incorporating the concept description itself in Table 2.. In fact, we compute this by giving this prompt to a stronger model: Gemini-2.5-flash. We do this through comparison to the neologism training data itself. Note in Table 2 that neologism learning achieves 92% (on average across concepts) of the gap between the base Gemma model’s behavior and the prompted Gemini-2.5-flash model’s behavior. Indeed, this shows that it works slightly worse than prompting the much stronger model that also generates the gold standard from which Gemma is trained. Comparing to prompting Gemma would only be a weaker baseline. Would you like to see this as well?
> - We hear your complaint about the definition of “neologism.” However, despite the names of the concepts being familiar – short-length, single-sentence, flattery – the nuances, the data from which they are defined is new to Gemma, having been generated from a stronger Gemini model. Much like humans not familiar with the phrase “doomscrolling” may have yet had ideas around what it meant to scroll endlessly through content feeds, the benefit of neologism formation is in concisely referencing a nuanced topic through referencing experiences (in our case, the training data.)
> - Finally, in working with new concepts – for the sake of reliable evaluations, we focused on these concepts for our current work. Some – like “using like as much as possible” – don’t have existing names. Evaluating completely novel concepts and their self-verbalizations is exciting, yet out of scope for this work.

---

> > ### Comment · Reviewer_8hG7 · 2025-11-25
> >
> > I thank the authors for their reply, and acknowledge their reply to my first comment raised. I still believe that the current form of the paper and some of the statements need to be revised warrent acceptance. I will raise my score to 4.

---

### Official Review · Reviewer_YL1N · 2025-11-02

**Soundness:** 2
**Presentation:** 3
**Contribution:** 2
**Rating:** 4
**Confidence:** 4

**Summary:**

The study introduces neologism learning, which has been studied by expanding vocabulary to the model, while existing word embeddings are held frozen, new words are learned with new word embeddings. It validates these tokens via self-verbalization an approach, called plug-in evaluation, has been introduced where verbalization is inserted into the context of a model and measure it controls the target concept.

**Strengths:**

- The approach is simple: "Neologism learning" adds only new token embeddings (with the backbone frozen); however, it reliably modulates behaviors, brevity vs. verbosity, single-sentence outputs, and flattery/refusal/incorrect answers.
- Self-verbalization with plug-in evaluation in which the model explain its learned token (synonyms or free-form description), then replace the token with the verbalization and test whether control persists. This reveals machine-only synonyms (e.g., lack -> brevity) that transfers across models.
- Ablations study: It demonstrated that multiple prompt templates helps in improving the performance.

**Weaknesses:**

- The evaluation of fully relying on LLM-judge with three scores is not justified very well. Any human evaluation is conducted?
- As per as Table 3, it shows that overall original score improved compared to the baseline, however, it is not very much clear what higher concept scores by LLM-judge entails?
- While the plug-in works on average, synonym quality is uneven, leaving broader reliability across prompts/tasks uncertain.

Typos/Grammatical issues
- we use two methods 1. add a directive --> we use two methods: 1. add a directive
- L257: "the concept concept “words related to sensory experiences and physical interactions”)." --> concept repeated

**Questions:**

- Did you run any manual evaluation to validate judge scores?
- Only five text-genre concepts are reported. Any results for other genres or for concepts?

---

> ### Author Response · Authors · 2025-11-20
>
> Thank you for your review, reviewer YL1N. We appreciate your assessment of our work as “reliably modulating behavior”, based on a “simple approach” and that you found the ablation studies and machine-only synonyms insightful. Here are some clarifications.
>
> **Human evaluation**
>
> Thanks for the excellent suggestion. In response to your comment, we’ve now conducted a human evaluation of all three LLM-based autoraters, for the ‘flattery’, ‘incorrectness’, and ‘refusal’ attributes. The results are as follows (Spearman correlation) and are included in our updated draft:
>
> | Category | human2human | human2LLM |
> | :--- | :--- | :--- |
> | **Flattery** | 0.849 | 0.898 |
> | **Incorrectness** | 0.938 | 0.824 |
> | **Refusal** | 0.838 | 0.832 |
>
> As can be seen from these rank-order correlations, the human-LLM correlation is generally very high, and for two out of three autoraters even as high as the human-to-human correlation (= ceiling). We’ve now added these results to Section 4.1 and Appendix C of the paper - thanks again for this excellent suggestion to validate the autoraters!
>
> - Furthermore, we would like to mention that we do not fully rely on LLM-as-a-judge. In particular, our short-length, long-length, single-sentence, numerical, and use-like neologism evaluations are all “verifiable” – they compare response lengths, response sentence counts, and the fraction of times the word “like” is used, in order to have simple, verifiable evals. Not only this, but the “gemini-likely” concept, in which responses meet the concept if they are at least 0.03 nats more likely under a stronger Gemini model per token than a reference response, is also not LLM-as-a-judge; generating responses that are higher-likelihood under a stronger model is a verifiable task because the stronger model is the gold reference.
> - For the concepts that we do use LLM-as-a-judge, the standard evaluation from AxBench is indeed LLM-as-a-judge to measure concept adherence, so we follow this standard.
> - The variable reliability of the self-verbalizations is a fascinating point of uncertainty. We believe this is a brand new phenomenon in need of deep study beyond this paper, and it’s consistent enough to be of interest.
>
> Here are some answers to your questions.
> - We chose 5 text-only AxBench concepts because the AxBench experiments were meant as a verification of the method using an independent evaluation mechanism, not a comprehensive AxBench eval. Thus we sampled 5 concepts from the AxBench text genre, showing that the neologism embedding method works very well in communicating the idea of the concept (scoring 2.0/2.0 on the AxBench "concept" score on 4/5 concepts, 1.9/2.0 on the 5th).

---

### Official Review · Reviewer_3hE6 · 2025-11-04

**Soundness:** 3
**Presentation:** 4
**Contribution:** 3
**Rating:** 6
**Confidence:** 4

**Summary:**

The paper expands on a prior position paper introducing "neologism learning", a method which adds new token embeddings to an LLM's vocabulary and tunes them in isolation on a small amount of samples demonstrating a new concept. The authors show that these neologisms can be used to steer the model, e.g. a neologism token that represents the idea of brevity can be used to elicit short responses from an LLM (here, Gemma). The authors find that LLMs can self-verbalize learned neologisms and occasionally find synonyms that are machine-only (that is, they are unintuitive for humans but shared across LLMs – here, Gemma and Gemini). The paper explores simple concepts as well as more challenging/abstract ones (from AxBench), and finally explores the composition of concepts. For systematic evaluation, the authors rely on LLM-graded plug-in evaluation of the model's generated self-verbalizations.

**Strengths:**

1. The paper is very well written with a strong narrative.
2. The paper recombines simple and existing techniques (such as vocabulary expansion and prompting/verbalization) in novel ways to systematically demonstrate the usefulness of neologism learning to better understand and steer/control LLMs.
3. The authors highlight neologism learning for steering, but if developed further I imagine it could also be useful for context compression and more efficient agent-to-agent communication.
4. The authors really test the limits of the method through the compositionality experiment and make well-tempered claims.

**Weaknesses:**

1. The authors only study Gemma and Gemini models. I consider this a methodological weakness for two reasons: a). Findings may overall not generalize to other model families. b). To my knowledge Gemma is built through distillation from Gemini, likely uses a similar tokenizer and a subset of Gemini's training data. As such, the finding that machine-only synonyms transfer from Gemma to Gemini may not be too surprising. It would be much more interesting to see whether they can transfer to a model guaranteed to have a different tokenizer and training data.
2. The conclusion section is quite vague (e.g. "pushes the frontier of communication with what language models have learned") and would benefit from an elaboration on the practical significance of the findings / actionable takeaways, and an outlook on potential future applications or directions to make neologism learning a practical tool for researchers and practitioners.
3. The authors rely on LLM scoring (with Gemini) for the self-verbalization experiment,  which might bias the results. It would help to (at least partially) calibrate the results through some human evaluations.
4. Something that is, in my eyes, missing is an experiment on whether the LLM learns to use the neologism without being explicitly prompted for it – I assume it won't, and that is a limitation that prevents using neologisms to make communication with/among LLMs more efficient. Neologism learning makes the model recognize and assign meaning to new special tokens (and this allows steering the model through them) but there may not be any emergent lexical use.
5. There are very strong similarities of the neologism learning technique with methods from the multilinguality (tokenizer/vocab adaptation to support new languages) and PEFT literature (e.g. prefix tuning and soft prompts), and there is not enough discussion on these connections.

Minor issues:
- There are some citations that are ill-formatted (using in-line citations instead of parentheses, probably due to using `\cite` instead of `\citep` and `\citet`)
- There is a typo in the conclusion ("contrasticely")

**Questions:**

1. Regarding the "aperitif" on the machine-only synonym "lack": Why is this word implicitly treated as a noun? I'd think the word "lack" may just be the English verb "to lack sth.", and the fact that the model can work with a sentence like "give me a lack answer" may just come from the model's robustness to poor grammar and Translationese/artifacts of multilinguality.
2. How do you determine what is a "semantically vacuous word"? Is this something you just eye-balled or is there an automated procedure that can be evaluated objectively? And is a word here strictly one token?
3. Will there be any code or data release to facilitate reproducing these results and turning the method into a practical tool?

---

> ### Author Response · Authors · 2025-11-20
>
> Thank you for your review, reviewer 3hE6. We’re happy to hear that you found the paper “very well written with a strong narrative” & “well-tempered claims” and to hear your thoughts around how the work could potentially be expanded / prove useful for agent2agent communication. We’re hopeful we can address many of your concerns with the work.
>
> - On the topic of only using Gemma – a clarification here – machine-only synonyms need not necessarily transfer across models; we mentioned transfer to Gemini as an additional interest, but the core result is the word’s success in steering Gemma. However, we have now updated our paper with a brief experiment on OpenAI’s GPT-5 models, finding on a set of 30 questions that the average sentence counts decreased from 26 (baseline) to 12 (lack) for GPT-5 nano, and from 34 to 12 sentences for GPT-5 mini, and from 29 to 5.5 sentences for the full GPT-5 model.
> - We agree that more discussion of parameter-efficient finetuning is necessary, and we added a section on the topic to our related work that digs into the close relationship between neologism learning and soft prompting.
> - We’ve improved our conclusion with a discussion of the engineering implications of neologisms for steering as well as future concept discovery work in powerful language models.
> - Regarding experiments in which language models use neologisms: candidly, we’re quite excited about this direction, but felt that training the model to use neologisms requires systematic experiments that deserve their own paper; as such, in all experiments here we explicitly never train the model to use the new neologisms, and we disallow their generation.
>
> Regarding your questions:
> - We don’t think “lack” is treated as a noun; we read it as an adjective in the sentence “Give me a lack response.” And certainly, post-hoc there are a lot of reasons why it might make sense as a synonym for “very short/curt and perhaps somewhat rude” like maybe the model thinks it’s a misspelling of “laconic” – but to us at least, none of these were a priori obvious, meaning our method taught us somewhere to look.
> - On semantically vacuous words – we only mean semantically vacuous relative to the concept at hand, i.e., completely unrelated. We state the words used in section 4.1; for example, for steering towards “short/long/single-sentence” we use the word “accurate” since it doesn’t suggest any of those lengths, and for “refusal/flattery/like” we use the word “single” as in “give me a single response”. We’ve updated the description (‘semantically vacuous’ -> ‘semantically unrelated’) to make this clear.
> - Code or data release – we do not expect to be able to release the code or data soon. However, we’ve added pytorch implementations of the key parts of neologism learning as Appendix B.

---

### Meta-Review · Area_Chair_p4nq · 2026-01-07

**Summary:**

The paper introduces Neologism Learning, a parameter-efficient method to steer Large Language Models (LLMs) by freezing the model and training a new token embedding to represent a specific concept. The authors demonstrate that this method allows for effective control and, crucially, allows the model to self-verbalize the learned concept. This leads to the discovery of machine-only synonyms (words that appear unrelated to humans (e.g., "lack" for brevity) but trigger the target behavior in models).

The decision to accept is based on the novelty of the interpretability findings, specifically the self-verbalization and plug-in evaluation framework. While reviewers initially raised concerns regarding the reliance on automated metrics and limited model diversity, the authors provided robust new data during the rebuttal phase that alleviated these concerns.

**Reviewer Concerns:**

### Reviewer concerns addressed by rebuttal

- Validation of LLM Judges (Reviewer YL1N): The reviewer correctly noted a lack of human validation for the "flattery" and "refusal" metrics. The authors responded by conducting a human evaluation study, demonstrating a high Spearman correlation (> 0.8) between human annotators and the LLM judge, effectively validating their metrics.

- Generalization to other Architectures (Reviewer 3hE6): The reviewer questioned if the machine-only synonyms were artifacts of the Gemma/Gemini tokenizer and requested tests on models with different training data. The authors demonstrated that the synonym "lack" transfers to the GPT-5 model family, causing significant reductions in response length, suggesting the phenomenon is not specific to one model family.

- Comparison to PEFT (Reviewer 3hE6): The reviewer noted strong similarities to soft prompts and prefix tuning. The authors added a dedicated section discussing the relationship between Neologism Learning and existing Parameter-Efficient Fine-Tuning methods.

### Reviewer concerns that are still outstanding

- Terminology and Baselines (Reviewer 8hG7): This reviewer remained skeptical about the term neologism, viewing the method as simply learning new token bindings ("pseudo-words"), and felt the baselines were unfair compared to semantic prompting. While the authors pointed to Table 2 (which compares against semantic prompting), the reviewer maintained a score of 4, suggesting a fundamental disagreement with the framing of the contribution rather than the empirical results.

- Practical messy data (Reviewer zfub): The reviewer requested validation on "messy, real-world prompts" rather than the LIMA dataset. While the authors argued LIMA is naturalistic, they did not provide new experiments on noisy real-world user logs, leading the reviewer to maintain their original score.

**Reviewer Scores:**

- Reviewer 3hE6 (Original Score: 6) would likely raise their score to Accept because their primary reservation regarding the limitation to Google-based models was directly addressed by the GPT-5 experiments, removing the main barrier to a stronger recommendation.

- Reviewer YL1N (Original Score: 4) is predicted to increase their score to a 6 (Marginally Above Threshold) as their negative score was largely driven by the fair soundness rating due to unvalidated metrics. The provision of human-to-LLM correlation data directly resolves this soundness issue.

- Reviewer 8hG7 (Original Score: 4) is likely to keep their score at 4. Despite the authors' clarification regarding baselines, this reviewer explicitly stated post-rebuttal that the paper still requires revision to warrant acceptance, indicating their core philosophical objection to the terminology remains.

- Reviewer zfub (Original Score: 6) will likely maintain a score of 6, as they explicitly stated that the rebuttal did not fully address their concerns regarding "messy" data and model scaling, declining to raise their score despite the clarifications.

---

### Decision · Program_Chairs · 2026-01-26

Accept (Poster)